# Self-supervised Learning of Motion Capture

**Hsiao-Yu Fish Tung [1], Hsiao-Wei Tung [2], Ersin Yumer [3], Katerina Fragkiadaki [1]**
[1] Carnegie Mellon University, Machine Learning Department
[2] University of Pittsburgh, Department of Electrical and Computer Engineering
[3] Adobe Research
{htung, katef}@cs.cmu.edu, hst11@pitt.edu, yumer@adobe.com

## Abstract

Current state-of-the-art solutions for motion capture from a single camera are optimization driven: they optimize the parameters of a 3D human model so that its re-projection matches measurements in the video (e.g. person segmentation, optical flow, keypoint detections etc.). Optimization models are susceptible to local minima. This has been the bottleneck that forced using clean green-screen like backgrounds at capture time, manual initialization, or switching to multiple cameras as input resource. In this work, we propose a learning based motion capture model for single camera input. Instead of optimizing mesh and skeleton parameters directly, our model optimizes neural network weights that predict 3D shape and skeleton configurations given a monocular RGB video. Our model is trained using a combination of strong supervision from synthetic data, and self-supervision from differentiable rendering of (a) skeletal keypoints, (b) dense 3D mesh motion, and (c) human-background segmentation, in an end-to-end framework. Empirically we show our model combines the best of both worlds of supervised learning and test-time optimization: supervised learning initializes the model parameters in the right regime, ensuring good pose and surface initialization at test time, without manual effort. Self-supervision by back-propagating through differentiable rendering allows (unsupervised) adaptation of the model to the test data, and offers much tighter fit than a pretrained fixed model. We show that the proposed model improves with experience and converges to low-error solutions where previous optimization methods fail.

## 1   Introduction

Detailed understanding of the human body and its motion from "in the wild" monocular setups would open the path to applications of automated gym and dancing teachers, rehabilitation guidance, patient monitoring and safer human-robot interactions. It would also impact the movie industry where character motion capture (MOCAP) and retargeting still requires tedious labor effort of artists to achieve the desired accuracy, or the use of expensive multi-camera setups and green-screen backgrounds.

Most current motion capture systems are optimization driven and cannot benefit from experience. Monocular motion capture systems optimize the parameters of a 3D human model to match measurements in the video (e.g., person segmentation, optical flow). Background clutter and optimization difficulties significantly impact tracking performance, leading prior work to use green screen-like backdrops [5] and careful initializations. Additionally, these methods cannot leverage the data generated by laborious manual processes involved in motion capture, to improve over time. This means

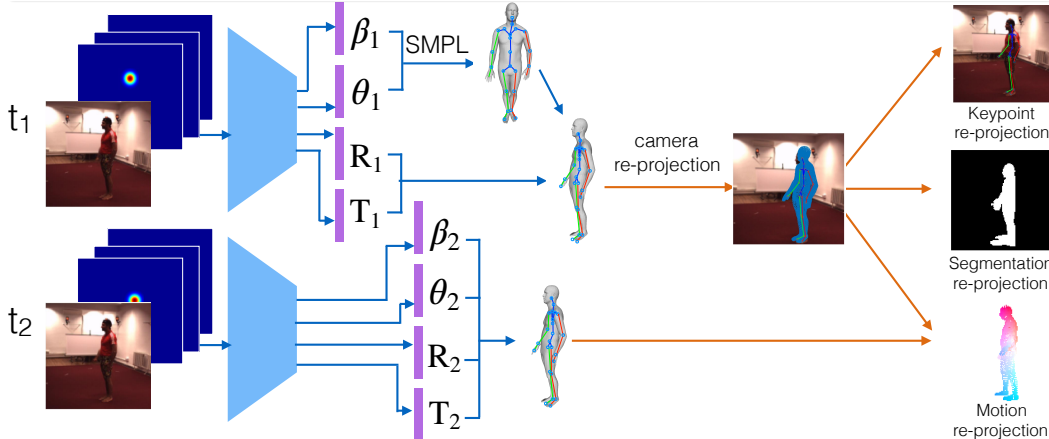

Figure 1: **Self-supervised learning of motion capture**. Given a video sequence and a set of 2D body joint heatmaps, our network predicts the body parameters for the SMPL 3D human mesh model [25]. Neural networks weights are pretrained using synthetic data and finetuned using self-supervised losses driven by differentiable keypoint, segmentation, and motion reprojection errors, against detected 2D keypoints, 2D segmentation and 2D optical flow, respectively. By finetuning its parameters at test time through self-supervised losses, the proposed model achieves significantly higher level of 3D reconstruction accuracy than pure supervised or pure optimization based models, which either do not adapt at test time, or cannot benefit from training data, respectively.

that each time a video needs to be processed, the optimization and manual efforts need to be repeated from scratch.

We propose a neural network model for motion capture in monocular videos, that learns to map an image sequence to a sequence of corresponding 3D meshes. The success of deep learning models lies in their supervision from large scale annotated datasets [14]. However, detailed 3D mesh annotations are tedious and time consuming to obtain, thus, large scale annotation of 3D human shapes in realistic video input is currently unavailable. Our work bypasses lack of 3D mesh annotations in real videos by combining strong supervision from large scale synthetic data of human rendered models, and *self-supervision* from 3D-to-2D differentiable rendering of 3D keypoints, motion and segmentation, and matching with corresponding detected quantities in 2D, in real monocular videos. Our self-supervision leverages recent advances in 2D body joint detection [37; 9], 2D figure-ground segmentation [22], and 2D optical flow [21], each learnt using strong supervision from real or synthetic datasets, such as, MPII [3], COCO [24], and flying chairs [15], respectively. Indeed, annotating 2D body joints is easier than annotating 3D joints or 3D meshes, while optical flow has proven to be easy to generalize from synthetic to real data. We show how state-of-the-art models of 2D joints, optical flow and 2D human segmentation can be used to infer dense 3D human structure in videos in the wild, that is hard to otherwise manually annotate. In contrast to previous optimization based motion capture works [8; 7], we use differentiable warping and differentiable camera projection for optical flow and segmentation losses, which allows our model to be trained end-to-end with standard back-propagation.

We use SMPL [25] as our dense human 3D mesh model. It consists of a fixed number of vertices and triangles with fixed topology, where the global pose is controlled by relative angles between body parts $\theta$, and the local shape is controlled by mesh surface parameters $\beta$. Given the pose and surface parameters, a dense mesh can be generated in an analytical (differentiable) form, which could then be globally rotated and translated to a desired location. The task of our model is to reverse-engineer the rendering process and predict the parameters of the SMPL model ($\theta$ and $\beta$), as well as the focal length, 3D rotations and 3D translations in each input frame, provided an image crop around a detected person.

Given 3D mesh predictions in two consecutive frames, we differentiably project the 3D motion vectors of the mesh vertices, and match them against estimated 2D optical flow vectors (Figure 1). Differentiable motion rendering and matching requires vertex visibility estimation, which we perform using ray casting integrated with our neural model for code acceleration. Similarly, in each frame, 3D keypoints are projected and their distances to corresponding detected 2D keypoints are penalized. Last but not the least, differentiable segmentation matching using Chamfer distances penalizes under and over fitting of the projected vertices against 2D segmentation of the human foreground. Note that

these re-projection errors are only on the shape rather than the texture by design, since our predicted 3D meshes are textureless.

We provide quantitative and qualitative results on 3D dense human shape tracking in SURREAL [35] and H3.6M [22] datasets. We compare against the corresponding optimization versions, where mesh parameters are directly optimized by minimizing our self-supervised losses, as well as against supervised models that do not use self-supervision at test time. Optimization baselines easily get stuck in local minima, and are very sensitive to initialization. In contrast, our learning-based MOCAP model relies on supervised pretraining (on synthetic data) to provide reasonable pose initialization at test time. Further, self-supervised adaptation achieves lower 3D reconstruction error than the pretrained, non-adapted model. Last, our ablation highlights the complementarity of the three proposed self-supervised losses.

## 2 Related Work

**3D Motion capture**    3D motion capture using multiple cameras (four or more) is a well studied problem where impressive results are achieved with existing methods [17]. However, motion capture from a single monocular camera is still an open problem even for skeleton-only capture/tracking. Since ambiguities and occlusions can be severe in monocular motion capture, most approaches rely on prior models of pose and motion. Earlier works considered linear motion models [16; 13]. Non-linear priors such as Gaussian process dynamical models [34], as well as twin Gaussian processes [6] have also been proposed, and shown to outperform their linear counterparts. Recently, Bogo et al. [7] presented a static image pose and 3D dense shape prediction model which works in two stages: first, a 3D human skeleton is predicted from the image, and then a parametric 3D shape is fit to the predicted skeleton using an optimization procedure, during which the skeleton remains unchanged. Instead, our work couples 3D skeleton and 3D mesh estimation in an end-to-end differentiable framework, via test-time adaptation.

**3D human pose estimation**    Earlier work on 3D pose estimation considered optimization methods and hard-coded anthropomorphic constraints (e.g., limb symmetry) to fight ambiguity during 2D-to-3D lifting [28]. Many recent works learn to regress to 3D human pose directly given an RGB image [27] using deep neural networks and large supervised training sets [22]. Many have explored 2D body pose as an intermediate representation [11; 38], or as an auxiliary task in a multi-task setting [32; 38; 39], where the abundance of labelled 2D pose training examples helps feature learning and complements limited 3D human pose supervision, which requires a Vicon system and thus is restricted to lab instrumented environments. Rogez and Schmid  [29] obtain large scale RGB to 3D pose synthetic annotations by rendering synthetic 3D human models against realistic backgrounds [29], a dataset also used in this work.

**Deep geometry learning**    Our differentiable renderer follows recent works that integrate deep learning and geometric inference [33]. Differentiable warping [23; 26] and backpropable camera projection [39; 38] have been used to learn 3D camera motion [40] and joint 3D camera and 3D object motion [30] in an end-to-end self-supervised fashion, minimizing a photometric loss. Garg et al. [18]learns a monocular depth predictor, supervised by photometric error, given a stereo image pair with known baseline as input. The work of [19] contributed a deep learning library with many geometric operations including a backpropable camera projection layer, similar to the one used in Yan et al. [39] and Wu et al. [38]'s cameras, as well as Garg et al.'s depth CNN [18].

## 3 Learning Motion Capture

The architecture of our network is shown in Figure 1. We use SMPL as the parametrized model of 3D human shape, introduced by Loper et al. [25]. SMPL is comprised of parameters that control the yaw, pitch and roll of body joints, and parameters that control deformation of the body skin surface. Let $\theta$, $\beta$ denote the joint angle and surface deformation parameters, respectively. Given these parameters, a fixed number ($n = 6890$) of 3D mesh vertex coordinates are obtained using the following analytical expression, where $\mathbf{X_i} \in \mathbb{R}^3$ stands for the 3D coordinates of the $i$th vertex in the mesh:

$$\mathbf{X}_i = \bar{\mathbf{X}}_i + \sum_m \beta_m \mathbf{s}_{m,i} + \sum_n (T_n(\theta) - T_n(\theta^*))\mathbf{p}_{n,i} \qquad (1)$$

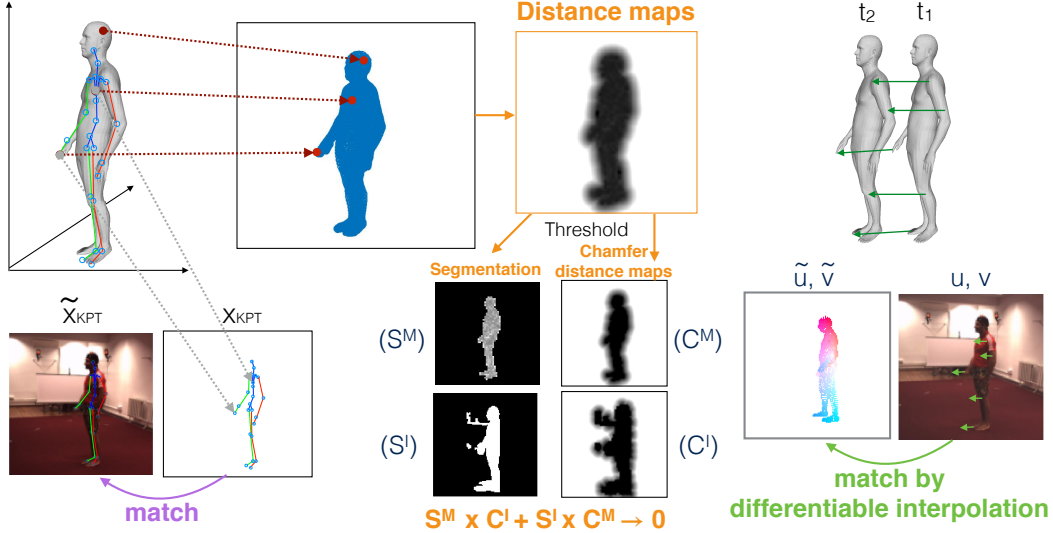

Figure 2: **Differentiable rendering** of body joints (left), segmentation (middle) and mesh vertex motion (right).

where $\bar{\mathbf{X}}_i \in \mathbb{R}^3$ is the nominal rest position of vertex $i$, $\beta_m$ is the blend coefficient for the skin surface blendshapes, $\mathbf{s}_{m,i} \in \mathbb{R}^3$ is the element corresponding to $i$th vertex of the $m$th skin surface blendshape, $\mathbf{p}_{n,i} \in \mathbb{R}^3$ is the element corresponding to $i$th vertex of the $n$th skeletal pose blendshape, $T_n(\theta)$ is a function that maps the $n$th pose blendshape to a vector of concatenated part relative rotation matrices, and $T_n(\theta^*)$ is the same for the rest pose $\theta^*$. Note the expression in Eq. 1 is differentiable.

Our model, given an image crop centered around a person detection, predicts parameters $\beta$ and $\theta$ of the SMPL 3D human mesh. Since annotations of 3D meshes are very tedious and time consuming to obtain, our model uses supervision from a large dataset of synthetic monocular videos, and self-supervision with a number of losses that rely on differentiable rendering of 3d keypoints, segmentation and vertex motion, and matching with their 2D equivalents. We detail supervision of our model below.

**Paired supervision from synthetic data**   We use the synthetic Surreal dataset [35] that contains monocular videos of human characters performing activities against 2D image backgrounds. The synthetic human characters have been generated using the SMPL model, and animated using Human H3.6M dataset [22]. Texture is generated by directly coloring the mesh vertices, without actual 3D cloth simulation. Since values for $\beta$ and $\theta$ are directly available in this dataset, we use them to pretrain the $\theta$ and $\beta$ branches of our network using a standard supervised regression loss.

### 3.1   Self-supervision through differentiable rendering

Self-supervision in our model is based on 3D-to-2D rendering and consistency checks against 2D estimates of keypoints, segmentation and optical flow. Self-supervision can be used at both train and test time, for adapting our model's weights to the statistics of the test set.

**Keypoint re-projection error**   Given a static image, predictions of 3D body joints of the depicted person should match, when projected, corresponding 2D keypoint detections. Such keypoint re-projection error has been used already in numerous previous works [38; 39]. Our model predicts a dense 3D mesh instead of a skeleton. We leverage the linear relationship that relates our 3D mesh vertices to 3D body joints:

$$\mathbf{X}_{kpt}{}^{\intercal} = \mathbf{A} \cdot \mathbf{X}^{\intercal} \qquad (2)$$

Let $\mathbf{X} \in \mathbb{R}^{4 \times n}$ denote the 3D coordinates of the mesh vertices in homogeneous coordinates (with a small abuse of notation since it is clear from the context), where $n$ the number of vertices. For estimating 3D-to-2D projection, our model further predicts focal length, rotation of the camera and

translation of the 3D mesh off the center of the image, in case the root node of the 3D mesh is not exactly placed at the center of the image crop. We do not predict translation in the $z$ direction (perpendicular to the image plane), as the predicted focal length accounts for scaling of the person figure. For rotation, we predict Euler rotation angles $\alpha, \beta, \gamma$ so that the 3D rotation of the camera reads $R = R^x(\alpha)R^y(\beta)R_t^z(\gamma)$, where $R^x(\theta)$ denotes rotation around the x-axis by angle $\theta$, here in homogeneous coordinates. The re-projection equation for the $k$th keypoint then reads:

$$x_{kpt}^k = P \cdot \left( R \cdot \mathbf{X}_{kpt}^k + T \right) \tag{3}$$

where $P = \text{diag}([f \quad f \quad 1 \quad 0]$ is the predicted camera projection matrix and $T = [T_x \quad T_y \quad 0 \quad 0]^T$ handles small perturbations in object centering. Keypoint reprojection error then reads:

$$\mathcal{L}^{\text{kpt}} = \|x_{kpt} - \tilde{x}_{kpt}\|_2^2, \tag{4}$$

and $\tilde{x}_{kpt}$ are ground-truth or detected 2D keypoints. Since 3D mesh vertices are related to $\beta, \theta$ predictions using Eq. 1, re-projection error minimization updates the neural parameters for $\beta, \theta$ estimation.

**Motion re-projection error** Given a pair of frames, 3D mesh vertex displacements from one frame to the next should match, when projected, corresponding 2D optical flow vectors, computed from the corresponding RGB frames. All Structure-from-Motion (SfM) methods exploit such motion re-projection error in one way or another: the estimated 3D pointcloud in time when projected should match 2D optical flow vectors in [2], or multiframe 2D point trajectories in [31]. Though previous SfM models use motion re-projection error to optimize 3D coordinates and camera parameters directly [2], here we use it to optimize neural network parameters, that predict such quantities, instead.

Motion re-projection error estimation requires visibility of the mesh vertices in each frame. We implement visibility inference through ray casting for each example and training iteration in Tensor Flow and integrate it with our neural network model, which accelerates by ten times execution time, as opposed to interfacing with raycasting in OpenGL. Vertex visibility inference *does not need to be differentiable*: it is used only to mask motion re-projection loss for invisible vertices. Since we are only interested in visibility rather than complex rendering functionality, ray casting boils down to detecting the first mesh facet to intersect with the straight line from the image projected position of the center of a facet to its 3D point. If the intercepted facet is the same as the one which the ray is cast from, we denote that facet as visible since there is no occluder between that facet and the image plane. We provide more details for the ray casting reasoning in the experiment section. Vertices that constructs these visible facet are treated as visible. Let $\mathbf{v}^i \in \{0, 1\}, i = 1 \cdots n$ denote visibilities of mesh vertices.

Given two consecutive frames $I_1, I_2$, let $\beta_1, \theta_1, R_1, T_1, \beta_2, \theta_2, R_2, T_2$ denote corresponding predictions from our model. We obtain corresponding 3D pointclouds, $\mathbf{X}_1^i = \begin{bmatrix} X_1^i \\ Y_1^i \\ Z_1^i \end{bmatrix}, i = 1 \cdots n$, and

$\mathbf{X}_2^i = \begin{bmatrix} X_2^i \\ Y_2^i \\ Z_2^i \end{bmatrix}, i = 1 \cdots n$ using Eq. 1. The 3D mesh vertices are mapped to corresponding pixel coordinates $(x_1^i, y_1^i), i = 1 \cdots n, (x_2^i, y_2^i), i = 1 \cdots n$, using the camera projection equation (Eq. 3). Thus the predicted 2D body flow resulting from the 3D motion of the corresponding meshes is $(u^i, v^i) = (x_2^i - x_1^i, y_2^i - y_1^i), i = 1 \cdots n$.

Let $\mathcal{OF} = (\tilde{u}, \tilde{v})$ denote the 2D optical flow field estimated with an optical flow method, such as the state-of-the-art deep neural flow of [21]. Let $\mathcal{OF}(x_1^i, y_1^i)$ denote the optical flow at a potentially subpixel location $x_1^i, y_1^i$, obtained from the pixel centered optical flow field $\mathcal{OF}$ through differentiable bilinear interpolation (differentiable warping) [23]. Then, the motion re-projection error reads:

$$\mathcal{L}^{\text{motion}} = \frac{1}{\mathbf{1}^T \mathbf{v}} \sum_i^n \mathbf{v}^i \left( \|u^i(x_1^i, y_1^i) - \tilde{u}(x_1^i, y_1^i)\|_1 + \|v^i(x_1^i, y_1^i) - \tilde{v}(x_1^i, y_1^i)\|_1 \right)$$

**Segmentation re-projection error**   Given a static image, the predicted 3D mesh for the depicted person should match, when projected, the corresponding 2D figure-ground segmentation mask. Numerous 3D shape reconstruction methods have used such segmentation consistency constraint [36; 2; 4], but again, in an optimization as opposed to learning framework.

Let $\mathcal{S}^I \in \{0, 1\}^{w \times h}$ denote the 2D figure-ground binary image segmentation, supplied by ground-truth, background subtraction or predicted by a figure-ground neural network segmenter [20]. Our segmentation re-projection loss measures how well the projected mesh mask fits the image segmentation $\mathcal{S}^I$ by penalizing non-overlapping pixels by the shortest distance to the projected model segmentation $\mathcal{S}^M = \{x_{2d}\}$. For this purpose Chamfer distance maps $\mathcal{C}^I$ for the image segmentation $\mathcal{S}^I$ and Chamfer distance maps $\mathcal{C}^M$ for the model projected segmentation $\mathcal{S}^M$ are calculated. The loss then reads:

$$\mathcal{L}^{\text{seg}} = \mathcal{S}^M \otimes \mathcal{C}^I + \mathcal{S}^I \otimes \mathcal{C}^M,$$

where $\otimes$ denotes pointwise multiplication. Both terms are necessary to prevent under of over coverage of the model segmentation over the image segmentation. For the loss to be differentiable we cannot use distance transform for efficient computation of Chamfer maps. Rather, we brute force its computation by calculating the shortest distance of each pixel to the model segmentation and the inverse. Let $x_{2d}^i, i \in 1 \cdots n$ denote the set of model projected vertex pixel coordinates and $x_{seg}^p, p \in 1 \cdots m$ denote the set of pixel centered coordinates that belong to the foreground of the 2D segmentation map $\mathcal{S}^I$:

$$\mathcal{L}^{\text{seg-proj}} = \underbrace{\sum_{i=1}^{n} \min_{p} \|x_{2d}^i - x_{seg}^p\|_2^2}_{\text{prevent over-coverage}} + \underbrace{\sum_{p}^{m} \min_{i} \|x_{seg}^p - x_{2d}^i\|_2^2}_{\text{prevent under-coverage}}. \tag{5}$$

The first term ensures the model projected segmentation is covered by the image segmentation, while the second term ensures that model projected segmentation covers well the image segmentation. To lower the memory requirements we use half of the image input resolution.

## 4   Experiments

We test our method on two datasets: Surreal [35] and H3.6M [22]. Surreal is currently the largest synthetic dataset for people in motion. It contains short monocular video clips depicting human characters performing daily activities. Ground-truth 3D human meshes are readily available. We split the dataset into train and test video sequences. Human3.6M (H3.6M) is the largest real video dataset with annotated 3D human skeletons. It contains videos of actors performing activities and provides annotations of body joint locations in 2D and 3D at every frame, recorded through a Vicon system. It does not provide dense 3D ground-truth though.

Our model is first trained using supervised skeleton and surface parameters in the training set of the Surreal dataset. Then, it is self-supervised using differentiable rendering and re-projection error minimization at two test sets, one in the Surreal dataset, and one in H3.6M. For self-supervision, we use ground-truth 2D keypoints and segmentations in both datasets, Surreal and H3.6M. The segmentation mask in Surreal is very accurate while in H3.6M is obtained using background subtraction and can be quite inaccurate, as you can see in Figure 4. Our model refines such initially inaccurate segmentation mask. The 2D optical flows for dense motion matching are obtained using FlowNet2.0 [21] in both datasets. We do not use any 3D ground-truth supervision in H3.6M as our goal is to demonstrate successful domain transfer of our model, from SURREAL to H3.6M. We measure the quality of the predicted 3D skeletons in both datasets, and we measure the quality of the predicted dense 3D meshes in Surreal, since only there it is available.

**Evaluation metrics**   Given predicted 3D body joint locations of $K = 32$ keypoints $\mathbf{X}_{kpt}^k, k = 1 \cdots K$ and corresponding ground-truth 3D joint locations $\tilde{\mathbf{X}}_{kpt}^k, k = 1 \cdots K$, we define the **per-joint error** of each example as $\frac{1}{K} \sum_{k=1}^{K} \|\mathbf{X}_{kpt}^k - \tilde{\mathbf{X}}_{kpt}^k\|_2$ similar to previous works [41]. We also define the **reconstruction error** of each example as the 3D per-joint error up to a 3D translation $T$ (3D

rotation should still be predicted correctly): $\min_T \frac{1}{K} \sum_{k=1}^{K} \|(\mathbf{X}_{kpt}^k + T) - (\tilde{\mathbf{X}}_{kpt}^k)\|_2$ We define the **surface error** of each example to be the per-joint error when considering all the vertices of the 3D mesh: $\frac{1}{n} \sum_{i=1}^{n} \|\mathbf{X}^i - \tilde{\mathbf{X}}^i\|_2$.

We compare our learning based model against two baselines: (1) *Pretrained*, a model that uses only supervised training from synthetic data, without self-supervised adaptation. This baseline is similar to the recent work of [12]. (2) *Direct optimization*, a model that uses our differentiable self-supervised losses, but instead of optimizing neural network weights, optimizes directly over body mesh parameters $(\theta, \beta)$, rotation $(R)$, translation $(T)$, and focal length $f$. We use standard gradient descent as our optimization method. We experiment with varying amount of supervision during initialization of our optimization baseline: random initialization, using ground-truth 3D translation, using ground-truth rotation and using ground-truth theta angles (to estimate the surface parameters).

Tables 1 and 2 show the results of our model and baselines for the different evaluation metrics. The learning based self-supervised model outperforms both the pretrained model, that does not exploit adaptation through differentiable rendering and consistency checks, as well as direct optimization baselines, sensitive to initialization mistakes.

**Ablation** In Figure 3 we show the 3D keypoint reconstruction error after self-supervised finetuning using different combinations of self-supervised losses. A model self-supervised by the keypoint re-projection error ($\mathcal{L}^{\text{kpt}}$) alone does worse than model using both keypoint and segmentation re-projection error ($\mathcal{L}^{\text{kpt}}+\mathcal{L}^{\text{seg}}$). Models trained using all three proposed losses (keypoint, segmentation and dense motion re-projection error ($\mathcal{L}^{\text{kpt}}+\mathcal{L}^{\text{seg}}+\mathcal{L}^{\text{motion}}$) outperformes the above two. This shows the complementarity and importance of all the proposed losses.

|  | **surface error** (mm) | **per-joint error** (mm) | **recon. error** (mm) |
|---|---|---|---|
| Optimization | 346.5 | 532.8 | 1320.1 |
| Optimization + $\tilde{R}$ | 301.1 | 222.0 | 294.9 |
| Optimization + $\tilde{R}$ + $\tilde{T}$ | 272.8 | 206.6 | 205.5 |
| Pretrained | 119.4 | 101.6 | 351.3 |
| Pretrained+Self-Sup | **74.5** | **64.4** | **203.9** |

Table 1: **3D mesh prediction results in Surreal [35].** The proposed model (pretrained+self-supervised) outperforms both optimization based alternatives, as well as pretrained models using supervised regression, that do not adapt to the test data. We use a superscript $\tilde{}$ to denote ground-truth information provided at initialization of our optimization based baseline.

|  | **per-joint error** (mm) | **recon. error** (mm) |
|---|---|---|
| Optimization | 562.4 | 883.1 |
| Pretrained | 125.6 | 303.5 |
| Pretrained+Self-Sup | **98.4** | **145.8** |

Table 2: **3D skeleton prediction results on H3.6M [22].** The proposed model (pretrained+self-supervised) outperforms both an optimization based baseline, as well as a pretrained model. Self-supervised learning through differentiable rendering allows our model to adapt effectively across domains (Surreal to H3.6M), while the fixed pretrained baseline cannot. Dense 3D surface ground-truth is not available and thus cannot be measured in H3.6M

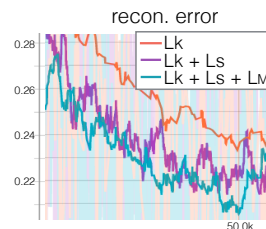

Figure 3: **3D reconstruction error during purely unsupervised finetuning** under different self-supervised losses. (Lk $\equiv$ $\mathcal{L}^{\text{kpt}}$: Keypoint re-projection error; LS$\equiv$ $\mathcal{L}^{\text{seg}}$: Segmentation re-projection error LM$\equiv$ $\mathcal{L}^{\text{motion}}$: Dense motion re-projection error ). All losses contribute to 3D error reduction.

**Discussion** We have shown that a combination of supervised pretraining and unsupervised adaptation is beneficial for accurate 3D mesh prediction. Learning based self-supervision combines the best of both worlds of supervised learning and test time optimization: supervised learning initializes the learning parameters in the right regime, ensuring good pose initialization at test time, without manual

effort. Self-supervision through differentiable rendering allows adaptation of the model to test data, thus allows much tighter fitting that a pretrained model with "frozen" weights at test time. Note that overfitting in that sense is desirable. We want our predicted 3D mesh to fit as tight as possible to our test set, and improve tracking accuracy with minimal human intervention.

**Implementation details** Our model architecture consists of 5 convolution blocks. Each block contains two convolutional layers with filter size $5 \times 5$ (stride 2) and $3 \times 3$ (stride 1), followed by

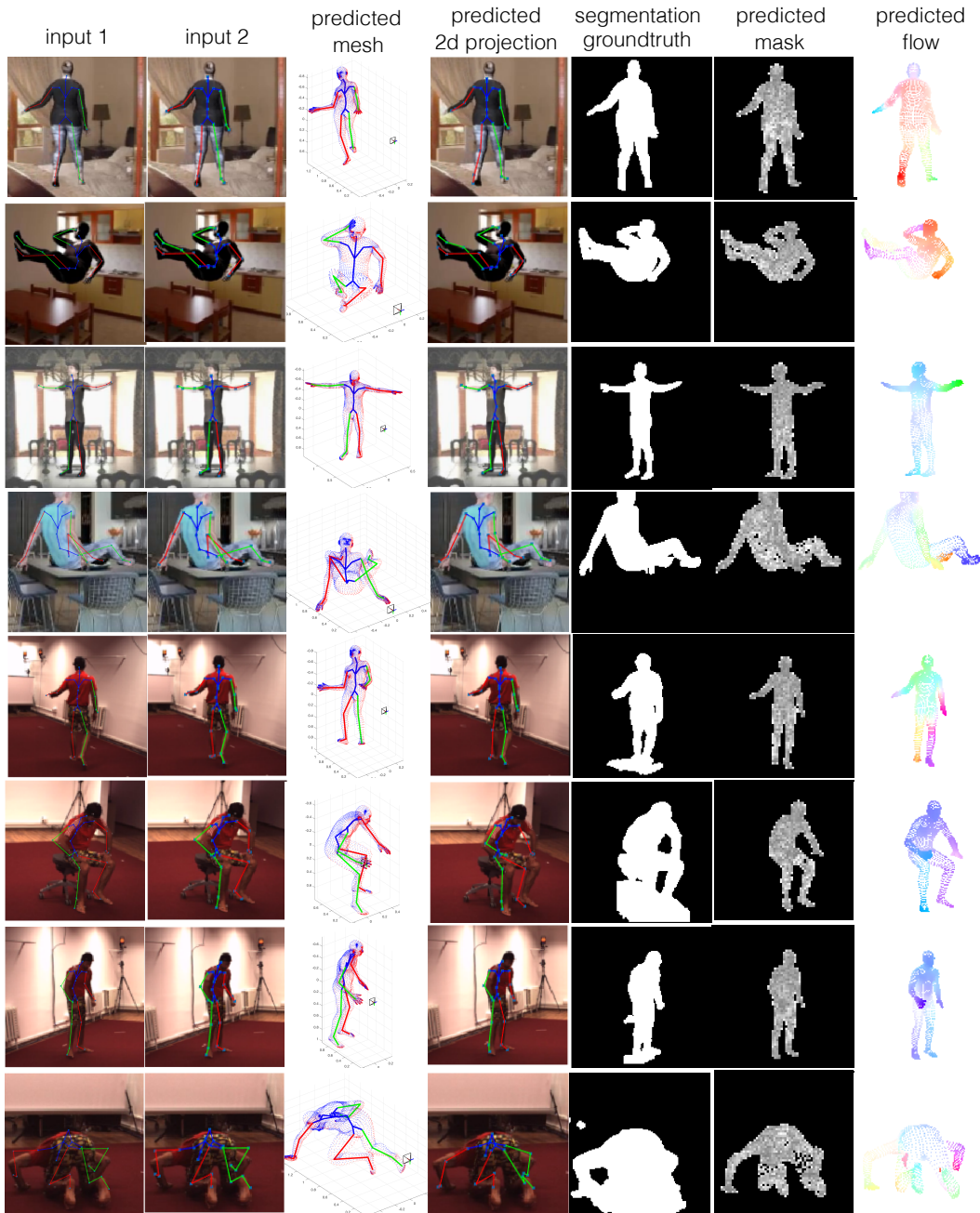

Figure 4: **Qualitative results of 3D mesh prediction**. In the top four rows, we show predictions in Surreal and in the bottom four from H3.6M. Our model handles bad segmentation input masks in H3.6M thanks to supervision from multiple rendering based losses. A byproduct of our 3D mesh model is improved 2D person segmentation (column 6).

batch normalization and leaky relu activation. The first block contains $64$ channels, and we double size after each block. On top of these blocks, we add 3 fully connected layers and shrink the size of the final layer to match our desired outputs. Input image to our model is $128 \times 128$. The model is trained with gradient descent optimizer with learning rate $0.0001$ and is implemented in Tensorflow v1.1.0 [1].

**Chamfer distance:** We obtain Chamfer distance map $\mathcal{C}^I$ for an input image frame $I$ using distance transform with seed the image figure-ground segmentation mask $\mathcal{S}^I$. This assigns to every pixel in $\mathcal{C}^I$ the minimum distance to a pixel on the mask foreground. Next, we describe the differentiable computation for $\mathcal{C}^M$ used in our method. Let $P = \{x_{2d}\}$ denote a set of pixel coordinates for the mesh's visible projected points. For each pixel location $p$, we compute the minimum distance between that pixel location and any pixel coordinate in $P$ and obtain a distance map $D \in \mathbb{R}^{w \times h}$. Next, we threshold the distance map $D$ to get the Chamfer distance map $\mathcal{C}^M$ and segmentation mask $\mathcal{S}^M$ where, for each pixel position $p$:

$$\mathcal{C}^M(p) = \max(0.5, D(p)) \tag{6}$$

$$\mathcal{S}^M(p) = \min(0.5, D(p)) + \delta(D(p) < 0.5) \cdot 0.5, \tag{7}$$

and $\delta(\cdot)$ is an indicator function.

**Ray casting:** We implemented a standard raycasting algorithm in TensorFlow to accelerate its computation. Let $r = (x, d)$ denote a casted ray, where $x$ is the point where the ray casts from and $d$ is a normalized vector for the shooting direction. In our case, all the rays cast from the center of the camera. For ease of explanation, we set $x$ at (0,0,0). A facet $f = (v_0, v_1, v_2)$, is determined as "hit" if it satisfies the following three conditions : (1) the facet is not parallel to the casted ray, (2) the facet is not behind the ray and (3) the ray passes through the triangle region formed by the three edges of the facet. Given a facet $f = (v_0, v_1, v_2)$, where $v_i$ denotes the $i$th vertex of the facet, the first condition is satisfied if the magnitude of the inner product between the ray cast direction $d$ and the surface normal of the facet $f$ is large than some threshold $\epsilon$. Here we set $\epsilon$ to be $1e - 8$. The second condition is satisfied if the inner product between the ray cast direction $d$ and the surface normal $N$, which is defined as the normalized cross product between $v_1 - v_0$ and $v_2 - v_0$, has the same sign as the inner product between $v_0$ on $N$. Finally, the last condition can be split into three sub-problems: given one of the edges on the facet, whether the ray casts on the same side as the facet or not. First, we find the intersecting point $p$ of the ray cast and the 2D plane expanded by the facet by the following equation:

$$p = x + d \cdot \frac{< N, v_0 >}{< N, d >}, \tag{8}$$

where $< \cdot, \cdot >$ denotes inner product. Given an edge formed by vertices $v_i$ and $v_j$, the ray casted is determined to fall on the same side of the facet if the cross product between edge $v_i - v_j$ and vector $p - v_j$ has the same sign as the surface normal vector $N$. We examine this condition on all of the three edges. If all the above conditions are satisfied, the facet is determined as hit by the ray cast. Among the hit facets, we choose the one with the minimum distance to the origin as the visible facet seen from the direction of the ray cast.

## 5 Conclusion

We have presented a learning based model for dense human 3D body tracking supervised by synthetic data and self-supervised by differentiable rendering of mesh motion, keypoints, and segmentation, and matching to their 2D equivalent quantities. We show that our model improves by using unlabelled video data, which is very valuable for motion capture where dense 3D ground-truth is hard to annotate. A clear direction for future work is iterative additive feedback [10] on the mesh parameters, for achieving higher 3D reconstruction accuracy, and allowing learning a residual free form deformation on top of the parametric SMPL model, again in a self-supervised manner. Extensions of our model beyond human 3D shape would allow neural agents to learn 3D with experience as human do, supervised solely by video motion.

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
