[Reviews · NeurIPS 2017]

Reviewer 1



This work addresses the problem of motion capture for monocular videos in the wild, where we are interested in estimating human poses in RGB videos. Instead of directly optimizing the parameters of a 3D human model to reduce the error measured by the metrics such as segmentation, optical flow, or the coordinates of key points, this paper proposes an end-to-end trainable framework to predict the parameters of the SMPL model. The optimization objective includes the projection error of key points, optical flow, and foreground-background segmentation. The model is tested on Surreal dataset and H3.6M dataset, compared with a pre-trained model and a direct optimization framework. [Strengths] - The paper proposes a set of promising objectives which can be utilized to fine-tune the models on real-world datasets, which are pre-trained on synthetic datasets. - Figures are well-drawn and do help readers understand the proposed framework. [Weakness] - This paper is poorly written. - The motivation, the contribution, and the intuition of this work can barely be understood based on the introduction. - Sharing the style of citations and bullet items is confusing. - Representing a translation vector with the notation $t^{i} = [x^{i}, y^{i}, z^{i} ] ^{T}$ is usually more preferred. - The experimental results are not convincing. - The descriptions of baseline models are unclear. - Comparing the performance of the methods which directly optimize the mesh parameters, rotation, translation, and focal length according to the metric provided (projection error) doesn't make sense since the objective is in a different domain of the measurement of the performance. - Comparing the performance of the model only pre-trained on synthetic data is unfair; instead, demonstrating that the proposed three projection errors are important is more preferred. In other words, providing the performance of the models pre-trained on synthetic data but fine-tuned on real-world datasets with different losses is necessary. - The reason of claiming that it is a supervised learning framework is unclear. In my opinion, the supervision signals are still labeled. [Reproducibility] The proposed framework is very simple and well explained with sufficient description of network parameters and optimization details. I believe it's trivial to reproduce the results. [Overall] In term of the proposed framework, this paper only shows the improvement gained of fine-tuning the model based on the proposed losses defined by the reprojection errors of key points, optical flow, and foreground-background segmentation. Taking into account that this work does show that fine-tuning the model pre-trained on synthetic datasets on real-world video clips improves the performance especially, it's still a convicting article. In sum, as far as I am concerned this work makes a contribution but is insufficient.

Reviewer 2



This paper proposes to do motion capture from monocular video using a neural network to directly predict 3D body configurations. The proposed model achieves state-of-the-art performance on benchmark datasets. Pros: - The overall proposed model with a network trained using a combination of supervision and self-supervised 3D/2D consistency checks appears to be novel. (Though I'm not very familiar with prior work in this area.) - The paper is well-written overall with good discussion of prior work on and motivation for the self-supervised losses used. The proposed techniques appear to be well-motivated with strong domain-specific knowledge of the motion capture problem. - The model achieves SotA results across two benchmark datasets and multiple evaluation metrics, and reasonable ablation studies are done to confirm the importance of each component of the self-supervised learning problem. Cons: - The paper focuses on a single application area which may not be of wide interest to the NIPS audience -- CVPR/ICCV/ECCV might be more appropriate venues. The novelty seems to be mainly application-driven rather than in any new learning or optimization techniques, by using existing domain-specific consistency checks to train a parametric model. - Some implementation details seem to be missing from the "Implementation details" section -- e.g., what is the input image resolution? When and how does spatial downsampling occur? (The stride of the convolutions is not mentioned.) Any data augmentation? Momentum? It would also be nice to see a figure or table summarizing the exact outputs of the model. - (Minor) Some of the losses include (non-squared) L2 norms -- is this correct? In standard linear regression the loss is the squared norm.

Reviewer 3



The paper proposes an end-to-end network for predicting the 3D human model, from monocular videos in the wild. Without having a direct supervision for the target (3D shape), the network is supervised using projections of the target in other spaces, with better results for videos in the wild (skeletal keypoints, dense motion and segmentation), for which they use state of the art implementations to obtain the labels. The functions used to build the projections from the 3D model are differentiable and are used to define the cost of the network. Because there might be multiple solutions in the 3D space that match the projections, the authors first calibrate the network with a synthetic dataset. The results are very good, showing an improvement of 5-6 times over previous implementations and 20%-50% over the pretrained solution. Regarding the results for components of the cost, the authors could add more plots (eg: Ls, Lk+Lm, Ls+Lm), in order to see what kind of information is missing from certain combinations. For future work they could consider finding other differentiable self-supervision cues, with higher state of the art results in those "missing information" zones (maybe tracking). The solution is not unsupervised because it clearly uses supervision from other state-of-the-art tools, with whom supervision the neural nets parameters that predict the 3D model are updated (maybe if the projections were obtained in an unsupervised manner). This is important in future work when the authors want to extend to other classes. But the formulas used to transform the 3D model into the 3 projections encloses only human supervision and is general enough to transfer to other classes of objects.